# Disease prediction with multi-omics and biomarkers empowers case–control genetic discoveries in the UK Biobank

Manik Garg [1,9], Marcin Karpinski[1,9], Dorota Matelska[1], Lawrence Middleton [1], Oliver S. Burren[1], Fengyuan Hu[1], Eleanor Wheeler [1], Katherine R. Smith [1], Margarete A. Fabre[1,2,3], Jonathan Mitchell[1], Amanda O'Neill[1,4], Euan A. Ashley [5], Andrew R. Harper [1,6], Quanli Wang[7], Ryan S. Dhindsa [7], Slavé Petrovski [1,8] ✉ & Dimitrios Vitsios [1] ✉

The emergence of biobank-level datasets offers new opportunities to discover novel biomarkers and develop predictive algorithms for human disease. Here, we present an ensemble machine-learning framework (machine learning with phenotype associations, MILTON) utilizing a range of biomarkers to predict 3,213 diseases in the UK Biobank. Leveraging the UK Biobank's longitudinal health record data, MILTON predicts incident disease cases undiagnosed at time of recruitment, largely outperforming available polygenic risk scores. We further demonstrate the utility of MILTON in augmenting genetic association analyses in a phenome-wide association study of 484,230 genome-sequenced samples, along with 46,327 samples with matched plasma proteomics data. This resulted in improved signals for 88 known ($P < 1 \times 10^{-8}$) gene–disease relationships alongside 182 gene–disease relationships that did not achieve genome-wide significance in the nonaugmented baseline cohorts. We validated these discoveries in the FinnGen biobank alongside two orthogonal machine-learning methods built for gene–disease prioritization. All extracted gene–disease associations and incident disease predictive biomarkers are publicly available (http://milton.public.cgr.astrazeneca.com).

Identifying individuals at high risk of developing disease is a priority for preventative medicine. Most traditional risk assessment tools rely on clinical parameters such as age, sex and family history, and a reduced set of basic biomarkers tailored to the disease under study[1–3]. However, these tools may not capture the full spectrum of biological processes that underlie complex diseases. The advent of large-scale biobanks that integrate electronic health records and multi-omics data—such as standard blood tests, proteomics and metabolomics—provide an unprecedented opportunity to discover novel biomarkers and collections of biomarkers to better predict disease onset.

The UK Biobank (UKB; https://www.ukbiobank.ac.uk) is one of the largest biobank cohorts, including health record and genetic sequencing data from half a million individuals aged between 40 and 69 at recruitment. There is a rich catalog of phenotype data for each participant, including continuously updated health record data, biometric measurements, lifestyle indicators, blood and urine biomarkers, and imaging. This biobank has revealed a variety of new genetic associations and candidate therapeutic targets[4–6]. Because of its scale, the UKB also offers an opportunity to identify combinations of biomarkers that may better predict disease onset than any single biomarker alone. For example, a recent paper used ~1,500 plasma protein measurements from over 50,000 UKB

**Fig. 1 | MILTON flowchart.** Individuals diagnosed with certain ICD10 codes in the UKB are herein referred to as 'cases' and all remaining individuals as 'controls' for that ICD10. Both cases and controls can have QVs, such as protein truncating variants, in a given gene. The objective of rare-variant collapsing analysis is to identify genes in which QVs are enriched in either cases or controls. Some controls may not yet be diagnosed with a given ICD10 code or are incorrectly classified. MILTON aims to identify these individuals by checking if they share similar biomarker profiles to known cases (represented by the shades of green). The predicted cases are eventually merged with the known cases to form an 'augmented case cohort' (ranging from 'L0' to 'L3'), which is analyzed along with a revised control set in an updated PheWAS on whole-genome sequencing (WGS) data.

participants to identify a handful of proteins that could accurately predict dementia even 10 years before diagnosis[7]. This suggests that a systematic approach across the phenotypes well-represented in the UKB could uncover many other accurate disease risk prediction models.

Beyond disease prediction and biomarker discovery, biomarker-based predictions of individuals with disease could also augment case–control genetic discovery analyses. More specifically, one limitation of most biobanks is that phenotype definitions often rely on billing codes and self-reporting, introducing potential misclassification of participants, missing data and variability in defining case–control cohorts[8]. Identifying individuals who may, in fact, be or become cases but are either not coded properly or have not yet been clinically diagnosed (that is, 'cryptic cases') remains an open challenge and opportunity.

Here, we introduce a systematic approach—MILTON—that aims to predict disease using commonly measured clinical biomarkers, plasma protein levels and other quantitative traits. We found these models predicted a range of different diseases with high accuracy. These results not only provide candidate biomarkers and combinations of biomarkers, but also clearly help augment and thus empower case–control analyses in some disease settings. Using these validated predictions, we performed augmented gene- and variant-level phenome-wide association studies (PheWASs)[8] on 3,213 phenotypes and 484,230 UKB genomes[9]. This identified several putative novel gene–phenotype associations that did not achieve significance in the baseline PheWAS from the same test cohort.

## Results

### Overview
Clinical biomarkers have a key role in diagnosing and evaluating many diseases, as they provide measurable indications of the presence

and/or severity of a condition. In the setting of PheWASs, they also provide an opportunity to identify cryptic or misclassified cases. Here, we introduce a machine-learning method, MILTON, that uses quantitative biomarkers to predict disease status for 3,213 disease phenotypes (Fig. 1 and Methods). MILTON first learns a disease-specific signature given a set of already diagnosed patients and then predicts putative novel cases among the original controls (Fig. 1 and Methods). The augmented cohorts are used to repeat rare-variant collapsing analysis[4] and compare the results against the baseline cohorts used to train the models (Fig. 1).

### Defining models based on sample collection and diagnosis dates
In the UKB, samples for biomarker measurement may have been collected as much as ~16.5 years before or 50 years after the corresponding individual was diagnosed with a disease (Fig. 2a). To determine the effect of this time-lag on predictive performance, we trained MILTON models on cases selected according to three different time-models: prognostic, diagnostic and time-agnostic (Fig. 2a), and five different time-lags (Methods). The prognostic model uses all individuals who received a diagnosis up to 10 years after biomarker sample collection; the diagnostic model uses all individuals who received a diagnosis up to 10 years before biomarker sample collection; and the time-agnostic model uses all diagnosed individuals for model development (Methods). We note that for the prognostic model, missing diagnoses may be attributed either to the disease not being present/diagnosed yet or to missing records in the biobank. The 10-year cut-off was selected as optimal after a biomarker sensitivity analysis on the effect of sample collection and diagnosis time-lag across 400 randomly selected ICD10 (International Classification of Diseases 10th revision[10]) codes (Methods and Supplementary Fig. 1).

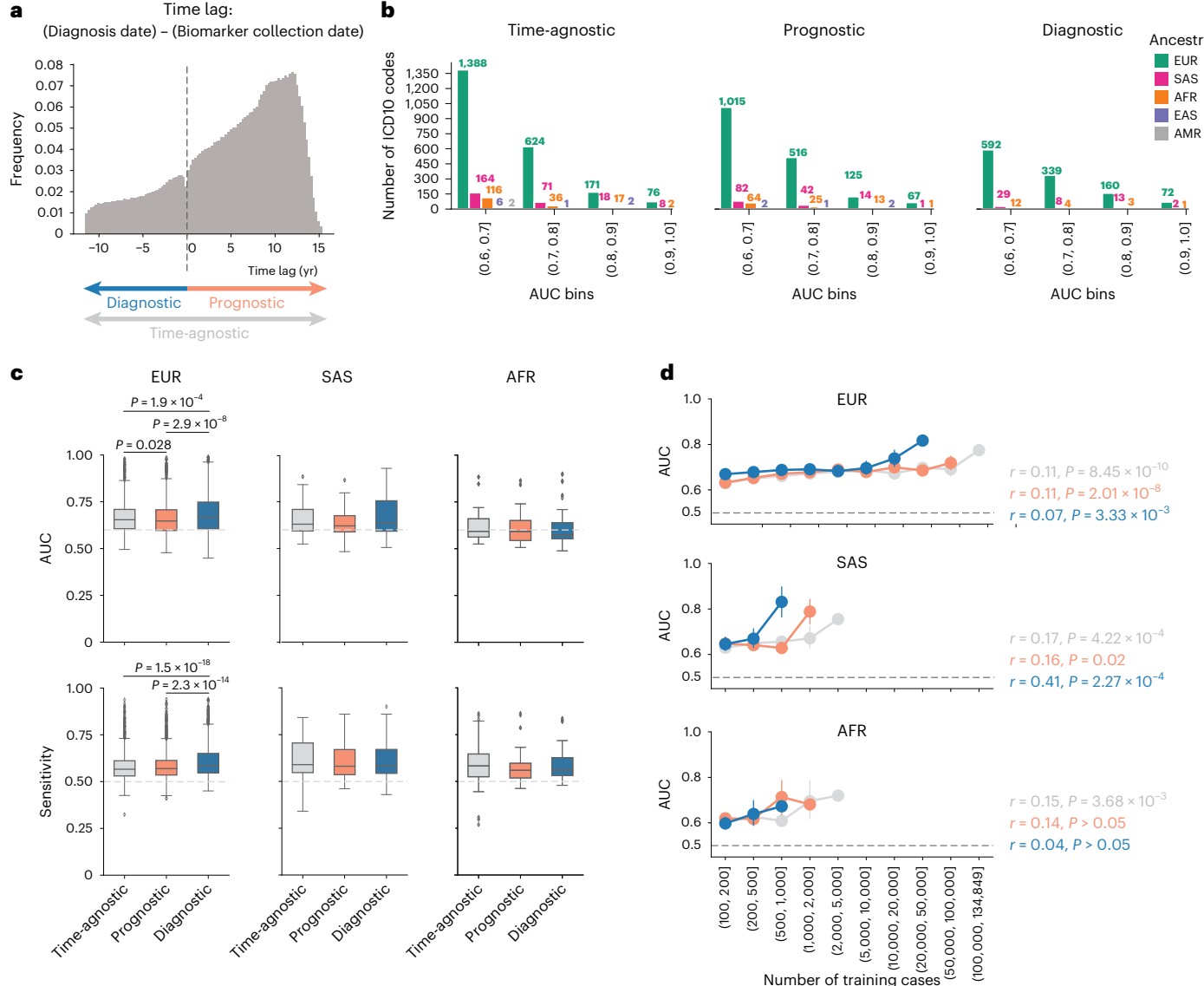

**Fig. 2 | MILTON time-models and phenome-wide performance across ancestries. a**, Schematic showing how different time-models are defined and the frequency of individuals that had biomarker sample collection certain years before or after diagnosis date. Diagnosis dates recorded in UKB fields 41280, 40000 or 40005 were taken for each individual (Methods). **b**, MILTON AUC performance across all ICD10 codes, five ancestries and three time-models. **c**, Comparison of median AUC and sensitivity performance of MILTON models across ten replicates trained on 1,466, 73 and 56 ICD10 codes under EUR, SAS and AFR ancestries, respectively, and different time-models. MWU,

two-sided *P* values are shown. Each box plot shows the median as center line, 25th percentile as lower box limit and 75th percentile as upper box limit, and whiskers extend to 25th percentile − 1.5 × interquartile range at the bottom and 75th percentile + 1.5 × interquartile range at the top; points denote outliers. **d**, Distribution of median AUC across ten replicates with increasing number of training cases per ICD10 code across different time-models and ancestries. Error-bar represents 95% confidence interval with center representing mean statistic. Pearson correlation coefficients (*r*) and two-sided *P* values (*P*) for each time-model are provided.

## MILTON disease prediction performance

In the first instance, MILTON was trained using 67 features including 30 blood biochemistry measures, 20 blood count measures, four urine assay measures, three spirometry measures, four body size measures, three blood pressure measures, sex, age and fasting time. After running MILTON across the phenome and multiple ancestries separately (Table 1), there were 3,200, 2,423 and 1,549 ICD10 codes based on time-agnostic, prognostic and diagnostic models, respectively, that satisfied our minimum set of robustness criteria (Supplementary Table 1b). Utilizing the area under the curve (AUC) to assess model performance, MILTON achieved AUC ≥ 0.7 for 1,091 ICD10 codes, AUC ≥ 0.8 for 384 ICD10 codes and AUC ≥ 0.9 for 121 ICD10 codes across all time-models and ancestries (Fig. 2b, Supplementary Table 2a–e and Supplementary Notes). MILTON achieved an AUC > 0.6 for more than half of

studied ICD10 codes per ICD10 chapter across 13 out of 18 chapters and all three time-models for individuals of European (EUR) ancestry (Supplementary Fig. 2a).

We found that diagnostic models generally had higher performance across 1,466 ICD10 codes, with results available for all three time-models in EUR ancestry participants (median AUC$_{diagnostic}$ versus AUC$_{prognostic}$: 0.668 versus 0.647, Mann–Whitney *U*-test (MWU) two-sided *P* = 2.86 × 10$^{-8}$; median sensitivity$_{diagnostic}$ versus sensitivity$_{prognostic}$: 0.586 versus 0.570, MWU two-sided *P* = 2.31 × 10$^{-14}$; Fig. 2c and Supplementary Table 2a–e). Overall, as the number of cases available for training per ICD10 increased, AUC, sensitivity and specificity remained stable for EUR and African (AFR) ancestries, whereas they increased for South Asian (SAS) diagnostic models (AUC: Pearson's *r* = 0.41, *P* = 2.27 × 10$^{-4}$; sensitivity: Pearson's *r* = 0.48,

**Table 1 | Summary of MILTON models' performance when trained on various feature sets and time-models**

| Time-model (abs(time-lag)) | Number of ICD10 codes | Features | Median AUC [0, 1] | Median sensitivity | Median specificity |
|---|---|---|---|---|---|
| Time-agnostic (no lag) | 151 | Disease-specific PRS+sex+age | 0.66±0.07 | 59.95±6.50% | 62.43±6.60% |
| Time-agnostic (no lag) | 151 | 67 traits | 0.71±0.12 | 61.93±10.95% | 69.44±11.98% |
| Prognostic (≤10 years) | 121 | Disease-specific PRS+sex+age | 0.65±0.06 | 59.73±5.99% | 62.05±6.21% |
| Prognostic (≤10 years) | 121 | 67 traits | 0.71±0.12 | 61.56±10.56% | 68.18±11.98% |
| Diagnostic (<10 years) | 82 | Disease-specific PRS+sex+age | 0.67±0.08 | 61.33±7.47% | 63.39±7.07% |
| Diagnostic (<10 years) | 82 | 67 traits | 0.75±0.14 | 65.65±12.22% | 71.49±13.09% |
| Time-agnostic (no lag) | 499 | 36 PRS+sex+age | 0.54±0.05 | 52.28±5.07% | 53.02±5.17% |
| Time-agnostic (no lag) | 499 | 67 traits | 0.64±0.09 | 56.46±7.34% | 63.79±9.10% |
| Prognostic (≤10 years) | 499 | 36 PRS+sex+age | 0.54±0.05 | 52.58±4.32% | 53.28±4.42% |
| Prognostic (≤10 years) | 499 | 67 traits | 0.65±0.10 | 57.87±7.65% | 65.20±9.47% |
| Diagnostic (<10 years) | 500 | 36 PRS+sex+age | 0.54±0.06 | 52.72±4.63% | 53.09±4.74% |
| Diagnostic (<10 years) | 500 | 67 traits | 0.67±0.10 | 58.71±8.56% | 65.76±9.49% |
| Time-agnostic (no lag) | 1,299 | 67 traits | 0.65±0.10 | 57.65±7.82% | 63.67±9.26% |
| Time-agnostic (no lag) | 1,299 | 3k proteins | 0.67±0.11 | 58.29±8.59% | 66.38±10.32% |
| Time-agnostic (no lag) | 1,299 | 3k proteins+67 traits | 0.68±0.11 | 58.62±8.76% | 66.59±10.33% |
| Prognostic (≤10 years) | 864 | 67 traits | 0.65±0.10 | 58.00±7.67% | 63.81±8.85% |
| Prognostic (≤10 years) | 864 | 3k proteins | 0.68±0.10 | 58.88±8.27% | 67.10±10.01% |
| Prognostic (≤10 years) | 864 | 3k proteins+67 traits | 0.69±0.11 | 59.20±8.59% | 67.36±10.18% |
| Diagnostic (≤10 years) | 435 | 67 traits | 0.66±0.11 | 58.79±8.89% | 64.07±9.46% |
| Diagnostic (≤10 years) | 435 | 3k proteins | 0.70±0.12 | 60.04±10.22% | 69.57±11.38% |
| Diagnostic (≤10 years) | 435 | 3k proteins+67 traits | 0.71±0.12 | 60.00±10.22% | 70.05±11.49% |

$P = 1.29 \times 10^{-5}$; specificity: Pearson's $r = 0.31$, $P = 0.0065$; Fig. 2d and Supplementary Fig. 2b).

**MILTON successfully predicts disease before onset**

To assess the effectiveness of MILTON in predicting genuine cases, we sought to determine whether individuals assigned a high case probability ($0.7 \leq P_{case} \leq 1$) by MILTON under the prognostic model were eventually diagnosed with those ICD10 codes in subsequent UKB phenotype refreshes. To investigate this, we trained MILTON models solely on cases diagnosed before 1 January 2018, and analyzed the predicted probability scores for cases diagnosed after this date (capped analysis, Fig. 3a).

Among 1,740 ICD10 codes with a minimum of 30 individuals diagnosed after 1 January 2018, and AUC ≥ 0.6 (Supplementary Tables 1b and 2f), 1,695 codes (97.41%) were significantly enriched in participants who had $P_{case} \geq 0.7$. This observation was supported by an odds ratio greater than 1 (Fisher's exact test one-sided $P < 0.05$) which persisted across prediction probability thresholds ≥ 0.3 (Fig. 3b and Supplementary Table 3). These results validate MILTON's ability to predict emerging cases from a pool of at the time undiagnosed participants, emphasizing its value for disease risk prediction and its potential for augmenting existing positive case labels for genetic association analyses.

**MILTON outperforms polygenic risk scores for disease prediction**

Polygenic risk scores (PRSs) are extensively researched to potentially aid disease diagnosis in clinics[11,12]. We compared the performance of MILTON models trained on 67 quantitative traits (Methods) with those trained on the respective disease-specific PRS[13] of a given phenotype, or all 36 standard PRSs available in the UKB. We used all 36 PRSs here to also consider the effects of non-disease-specific PRSs, such as PRSs for total cholesterol, PRSs for height and so on, on prediction of various diseases.

We observed that MILTON time-agnostic models trained on 67 quantitative traits significantly outperformed those trained on a single disease-specific PRS (including sex and age as covariates) for 111 out of 151 ICD10 codes (median $AUC_{67\,traits}$ versus $AUC_{disease-specific\,PRS}$: 0.71 versus 0.66, MWU two-sided $P = 2.71 \times 10^{-8}$; Table 1, Fig. 3c, Supplementary Fig. 3a and Supplementary Table 4a). We observed the same trend for both prognostic and diagnostic models (Supplementary Notes). Of note, PRSs for breast cancer (C50), melanoma (C43, D03) and prostate cancer (C61) performed better than the 67 quantitative traits in all three time-models (Supplementary Fig. 3a), likely due to the blood- and urine-based biomarkers used by MILTON carrying less predictive values for these solid cancers. In an additional analysis, we trained MILTON models including all 36 standard PRSs provided in the UKB and again observed that models trained on the 67 traits significantly outperformed those trained on PRSs for 499 randomly selected ICD10 codes (median $AUC_{67\,traits}$ versus $AUC_{all\,PRSs}$: 0.64 versus 0.54, MWU two-sided $P = 2.17 \times 10^{-82}$; Table 1, Fig. 3d, Supplementary Fig. 3c and Supplementary Table 4b).

**Plasma proteomics data improve performance for several diseases**

In addition to standard clinical biomarkers, the availability of other omics modalities provides additional features for predicting cases. Recently, the UKB Pharma Proteomics Project consortium profiled 2,923 plasma proteins in a subset of 49,736 UKB participants[14,15]. Using the EUR ancestry subset ($n = 46,327$) of the UKB cohort, we retrained MILTON incorporating the proteomics data, both in isolation and in combination with the other 67 biomarkers already analyzed (Supplementary Notes). This led to slightly improved overall performance (median $AUC_{3k\,proteins + 67\,traits}$ versus $AUC_{67\,traits}$: 0.68 versus 0.65, MWU two-sided $P = 3.24 \times 10^{-9}$; Table 1, Fig. 3e,f and Supplementary Table 4c–e), with 52 phenotypes having an AUC improvement of ≥ 0.1 (Fig. 3f).

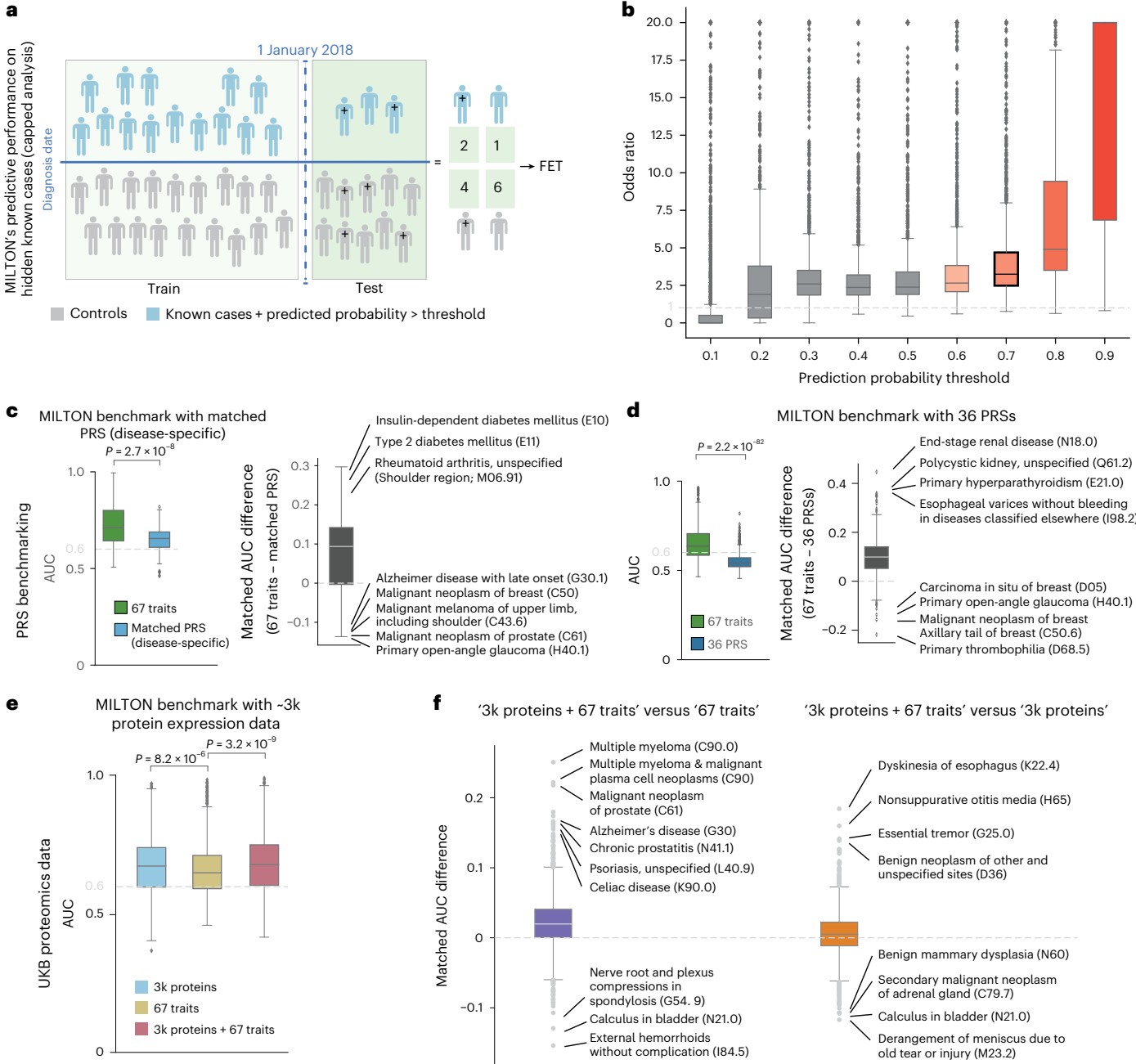

**Fig. 3 | MILTON validation and benchmarks with proteomics data and PRSs.**
**a**, Overview of capped analysis. Here, all individuals diagnosed until 1 January 2018 were used during model training and all individuals diagnosed thereafter were used as the test set for predictions. A 2 × 2 contingency table was constructed to capture whether known cases and controls were eventually correctly predicted by MILTON. **b**, Distribution of odds ratio obtained from Fisher's exact test (FET) in capped analysis on 1,748 ICD10 codes across multiple prediction probability thresholds, indicating the power of MILTON to predict known cases hidden from the training set. Results with predicted probability threshold ≥ 0.6 are filled with orange color and those corresponding to threshold = 0.7 are highlighted in black boundary. **c**, Performance comparison of MILTON time-agnostic models when trained on 67 traits versus disease-specific PRSs across 151 ICD10 codes. **d**, Box plots comparing the performance of MILTON

time-agnostic models when trained on 67 traits versus all 36 PRSs across 499 ICD10 codes. **e**, Performance comparison of MILTON time-agnostic models when trained on protein expression data + covariates ± 67 traits versus 67 traits across 1,574 ICD10 codes (Methods). **f**, AUC differences when MILTON is trained on different feature set combinations for 1,299 ICD10 codes (time-agnostic model). Left, $x$ axis represents median $AUC_{3k\,proteins+67\,traits}$ − median $AUC_{67\,traits}$ for matched ICD10 codes. Right, $x$ axis represents median $AUC_{3k\,proteins+67\,traits}$ − median $AUC_{3k\,proteins}$ for matched ICD10 codes. In **b**–**f**, each box plot shows median as center line, 25th percentile as lower box limit and 75th percentile as upper box limit; whiskers extend to 25th percentile − 1.5× interquartile range at the bottom and 75th percentile + 1.5× interquartile range at the top; points denote outliers. MWU, two-sided $P$ values are shown in **c**–**e**.

Several phenotypes benefited considerably from the inclusion of plasma proteomics data (Supplementary Fig. 3d and Supplementary Notes), including C90 (multiple myeloma and malignant plasma cell neoplasms), with AUC improving from 0.63 to 0.85. This was largely

driven by the addition of TNF receptor superfamily member 13B (TNFRSF13B or TACI) and member 17 (TNFRSF17 or BCMA) protein measurements[16–19] (Fig. 3f, Supplementary Fig. 4b and Supplementary Table 5f–h). Likewise, for C61 (prostate cancer), the AUC improved

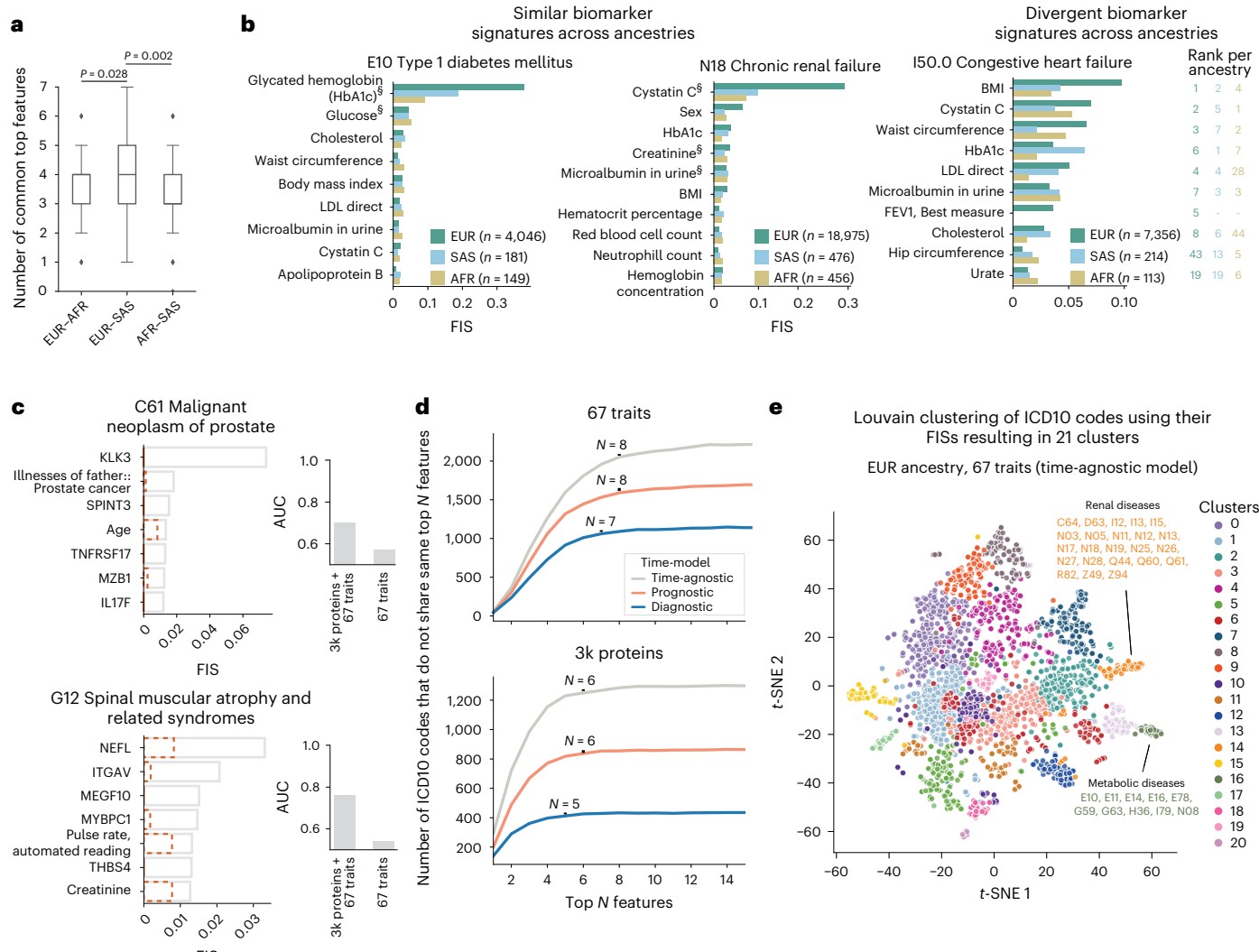

**Fig. 4 | Overview of most important biomarker features learnt by MILTON per ICD10 code for time-agnostic models. a**, Number of top seven biomarkers shared between each pair of ancestries for all 149 ICD10 codes with AUC > 0.6. MWU, two-sided *P* values are shown. No multiple testing correction was performed. Box plot shows median as center line, 25th percentile as lower box limit and 75th percentile as upper box limit; whiskers extend to 25th percentile − 1.5× interquartile range at the bottom and 75th percentile + 1.5 × interquartile range at the top; points denote outliers. **b**, Features with the highest FISs for E10 (type 1 diabetes mellitus), N18 (chronic renal failure) and I50.0 (congestive heart failure) for each ancestry. §Biomarkers that were also listed by an expert for given disease area[22]. LDL, low-density lipoprotein; FEV1, forced expiratory volume in 1 s. **c**, Top predictive

features for C61 and G12 when using UKB proteomics data to train MILTON (time-agnostic model). Dashed, orange bar plots indicate average FIS of corresponding feature across all ICD10 codes for time-agnostic model. Bar plots comparing AUC between models trained on proteomics data along with 67 traits versus 67 traits only are shown on the right. **d**, Number of ICD10 codes that do not share the top *N* features as a function of *N*, indicating a quasi-unique biomarker signature per disease, comprising *N* ≥ 7 features when models are trained on 67 traits only and *N* ≥ 5 features when models are trained on proteomics data only. **e**, The *t*-distributed stochastic neighbor embedding (*t*-SNE) projection of diseases across the phenome based on their MILTON-derived FISs. Each point corresponds to an ICD10 code, colored by Louvain clustering.

from 0.54 to 0.76 due to the addition of the known prostate cancer antigen (KLK3; Figs. 3f and 4c). Other examples included G12 (spinal muscular atrophy and related syndromes), with AUC improving from 0.57 to 0.70, likely due to the addition of neurofilament light chain (NEFL) protein[20,21] (Fig. 4c). These results highlight the value of adding additional features for certain diseases, and, in particular, the power of proteomics data in boosting predictions.

**MILTON identifies predictive features and disease clusters**

MILTON enables us to infer the importance of each feature in defining disease phenotypes (Supplementary Figs. 4, 5 and 6) as well as their concordance across ancestries (Fig. 4a and Supplementary Notes). We observed that MILTON assigned high feature importance scores (FISs) to at least one of the listed biomarkers for the corresponding disease chapter[22] (Supplementary Fig. 6 and Supplementary Table 5a–e). For example,

glycated hemoglobin (HbA1c) and glucose ranked as the top two features for type 1 diabetes mellitus (E10: AUC$_{all three ancestries, time-agnostic model}$ = 0.93 ± 0.04; Fig. 4b), which is expected as they are used for the clinical diagnosis of diabetes. Cystatin C, microalbumin in urine and creatinine ranked within the top five in chronic renal failure, and sex was one of the top features for predicting chronic renal failure across all ancestries[23–26] (ICD10: N18; Fig. 4b). This indicates that MILTON can distinguish between male- and female-specific cut-offs across the different biomarkers, as in certain diseases reference ranges for some biomarkers may be sex-specific.

We looked at the top features for those ICD10 codes where MILTON showed improved performance upon addition of proteomics data (Fig. 4c and Supplementary Fig. 4a,b), and, as a positive validation, we confirmed that MILTON ranks biomarkers known to be associated with certain diseases as top predictive features[27–30] (Fig. 4b,c and

**Table 2 | Number of unique gene–ICD10 associations ($P < 1 \times 10^{-8}$)**

| | Category | Number of unique associations | Number of unique ICD10 codes | Number of unique genes |
|---|---|---|---|---|
| ICD10 codes collapsed to three-character level | Known binary (with same direction of effect) | 236 | 116 | 82 |
| ICD10 codes collapsed to three-character level | Known quant | 1,047 | 397 | 70 |
| ICD10 codes collapsed to three-character level | Putative novel | 182 | 145 | 57 |
| All ICD10 codes | Known binary (with same direction of effect) | 506 | 293 | 82 |
| All ICD10 codes | Known quant | 2,168 | 1,057 | 70 |
| All ICD10 codes | Putative novel | 231 | 200 | 57 |

The gene–ICD10 association with the lowest $P$ value across all ten nonsynonymous QV models, three time-models and four augmented cohorts was reported.

Supplementary Fig. 4b). We also explored the number of features required to uniquely characterize a phenotype, identifying the top seven or eight most important features per disease as sufficient to provide a close to unique signature for each disease (based on <5% change in the number of ICD10 codes that share top $N$ and top $N - 1$ features; Fig. 4d). Similarly, 5–6 top features out of ~3,000 proteins + clinical covariates may be sufficient to distinguish one ICD10 from another (Fig. 4d). Eventually, we generated groups of phenotypes enriched for similar biomarker profiles (Fig. 4e and Supplementary Table 6a) which allow us to explore comorbidity profiles across patient cohorts (Supplementary Table 6b and Supplementary Notes). We have made all top predictive features per disease and the disease clusters available in our public portal (http://milton.public.cgr.astrazeneca.com).

### PheWAS on MILTON-augmented cohorts reveals putative novel signals

MILTON's power on disease risk prediction opens an additional potential: to augment existing positive case labels for genetic association analyses. We extracted MILTON-augmented cohorts ('L0'–'L3'; Supplementary Fig. 7a, Methods and Supplementary Notes) for 2,371 ICD10 codes with AUC > 0.6 for EUR ancestry, 271 for SAS ancestry, 179 for AFR ancestry, nine for East Asian (EAS) ancestry and two for American (AMR) ancestry. Here, MILTON-augmented cohorts contain all known cases along with an increasing number of MILTON-predicted cases as we go from L0 (conservative) to L3 (more inclusive) predictions (Methods). Rare-variant collapsing analysis on whole genomes derived from these augmented cohorts from EUR ancestry resulted in 2,905 significant gene–ICD10 associations ($P_{MILTON} < 1 \times 10^{-8}$) between 1,207 ICD10 codes and 165 genes, 99.93% of which have lowest $P$ values in nonsynonymous qualifying variant (QV)[4] models (Supplementary Table 7a).

To benchmark the MILTON results with a reference dataset, we performed binary PheWAS analysis on the baseline cohorts for each ICD10 code and recovered 236 out of 270 gene–disease associations from baseline analyses in the augmented cohorts with the same direction of effect ($P_{MILTON}$, $P_{baseline} < 1 \times 10^{-8}$), labeling them as 'known binary' (Table 2, Supplementary Fig. 7b and Supplementary Notes). For several known signals, evaluated as positive controls[31–37], MILTON achieved enhanced PheWAS results (Fig. 5a and Supplementary Notes). When characterizing putative novel signals, we sought to detect signals that may be reflecting correlation with a biomarker rather than independent disease association. To this end, we consider any genes achieving genome-wide significance with a biomarker in a baseline quantitative PheWAS and assess whether they rank among the top predictors for a given phenotype in the MILTON cohorts (Methods and Supplementary Notes). These accounted for 1,047 significant gene–disease associations which we do not consider as putative novel (Supplementary Notes).

Finally, we labeled the remaining 182 gene–disease associations ($P_{MILTON} < 1 \times 10^{-8}$, $P_{baseline} > 1 \times 10^{-8}$ and $P_{quantitative PheWAS for FIZ > 1.2} > 1 \times 10^{-8}$; Table 2) as 'putative novel' (Fig. 5b,c and Supplementary Notes). FIZ refers to the feature importance $Z$-score of highly predictive biomarkers per ICD10 code. To characterize an association as 'putative novel', we require that for any biomarker achieving FIZ > 1.2 during MILTON training (for a particular ICD10 code), the respective quantitative PheWAS associations are non-significant ($P_{quantitative PheWAS} > 1 \times 10^{-8}$). This filtering is applied to ensure any putative novel signals do not reflect correlations with biomarkers but instead represent independent disease associations (Supplementary Notes). Overall, MILTON reported 231 putative novel hits across all ICD10 codes (Table 2), 76.37% of which also reached nominal significance ($P < 0.05$) in baseline PheWASs. These putative novel hits achieved significantly lower $P$ values in baseline cohorts across all phenotypes compared with randomly selected genes of the same size (Supplementary Fig. 8). As a positive control observation, chapters XXI (health services) and XVIII (lab findings) were ranked as the top chapters with putative novel associations as they inherently capture individuals with atypical biomarker levels, which is the primary focus of MILTON (Supplementary Fig. 7g).

We then applied PheWAS to non-EUR ancestries, which represent a smaller proportion of UKB, and observed genome-wide significant associations only in SAS ancestry (eight associations between three genes and eight ICD10 codes; Supplementary Table 7b). One gene–phenotype association was shared between EUR and SAS, which was the hemoglobin subunit beta (HBB)–D56 thalassemia baseline association. After combining binary PheWAS results corresponding to protein truncating variants from all ancestries using Cochran–Mantel–Haenszel test, we identified nine associations in MILTON-augmented cohorts that achieved genome-wide significance ($P < 1 \times 10^{-8}$) in pan-ancestry analysis but had $P > 1 \times 10^{-8}$ in EUR ancestry-specific analysis (Supplementary Fig. 9 and Supplementary Table 7c). An example is apolipoprotein B (APOB)–E05 thyrotoxicosis (hyperthyroidism) association which achieved $P = 9.02 \times 10^{-9}$ in pan-ancestry analysis but had borderline significance ($P = 1.77 \times 10^{-8}$) in EUR-specific analysis. An intronic variant, rs12720793, in APOB gene is also associated with thyrotoxicosis in the FinnGen Biobank ($P < 0.008$). Conversely, we identified 33 associations in baseline cohorts and 274 associations in augmented cohorts that were genome-wide significant in EUR-specific analysis but not in pan-ancestry analysis (Supplementary Fig. 9 and Supplementary Table 7c).

### Exome-wide association studies on MILTON-augmented cohorts

To assess whether there is added benefit from using MILTON extended cohorts for common variant association studies, we applied MILTON to variant-level enrichment analysis (exome-wide association study (ExWAS)) across all phenotypes ($n = 2,259$ with AUC > 0.6), employing our ExWAS pipeline from previous work[4]. Based on the allelic ExWAS model, we observed that the odds ratio for >99% of variant–ICD10 associations ($P < 0.05$) remained consistent in the direction of effect between MILTON cohorts and baseline cohorts (Supplementary Fig. 12a). The percentage of associations with boosted $P$ values compared with baseline increased from L0 to L3 cohorts (53.94% to 65.25%; Supplementary Fig. 12b), as did the number of genome-wide

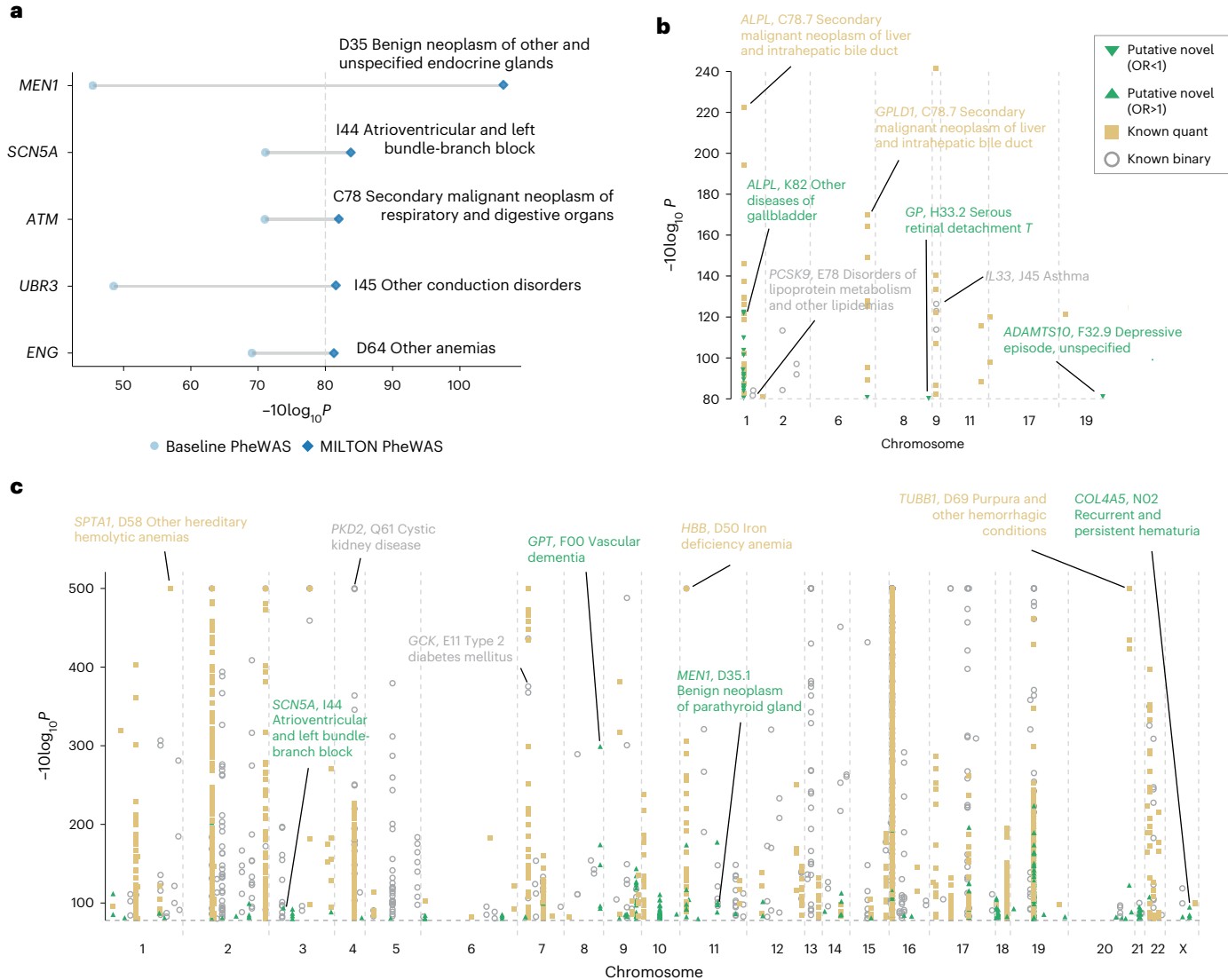

**Fig. 5 | PheWAS results on MILTON-augmented cohorts, based on whole-genome sequencing data, and stratification across known and putative novel hits. a**, Examples of known gene–disease associations from literature that reached genome-wide significance via MILTON. **b,c** Manhattan plots showing the distribution of gene–ICD10 associations with odds ratio (OR) < 1 (**b**) and OR > 1 (**c**) across different chromosome positions. For **a**–**c**, FET was used to calculate *P* values and odds ratios (two-sided, unadjusted).

significant associations (Supplementary Fig. 12c). Overall, we recovered with MILTON 6,321 out of 8,013 variant-level baseline associations (78.9%; $P < 1 \times 10^{-8}$) and observed 15,490 'known quantitative' along with 9,882 putative novel associations (Fig. 6c, Supplementary Fig. 13 and Supplementary Table 7d). Among the 9,882 putative novel ExWAS associations, 61.94% ($P = 6,121$) achieved $P < 0.05$ in baseline cohorts (Fig. 6c). For further granularity, we also performed common variant genome-wide association studies (GWASs) on 20 ICD10 codes and for each of the MILTON-augmented as well as baseline cohorts, again confirming that we achieve a good enrichment of true cases in the MILTON cohorts (Supplementary Table 9b, Methods and Supplementary Notes).

**Validation with FinnGen Biobank**
We aimed to identify support for the MILTON ExWAS-based putative novel hits in external datasets, and specifically in the FinnGen Biobank[38], which has variant-level enrichment results available (Supplementary Notes). Among all MILTON putative novel ExWAS associations that can be tested in FinnGen, 54.76% ($n = 2,002$) achieved $P < 0.05$ in FinnGen[38] release 10 (Fig. 6d and Supplementary Table 9a). For reference, among

the genome-wide significant hits inferred in baseline PheWASs, 88.76% ($n = 4,525$) had supporting evidence in FinnGen ($P < 0.05$; Fig. 6d).

For validation of common variant–ICD10 associations obtained from GWASs, we investigated 14 ICD10 codes that can be mapped to summary statistics results in FinnGen Biobank[38] release 11 (Supplementary Table 8). Among the unique baseline associations ($P < 1 \times 10^{-8}$) that could be tested in FinnGen, 93.10% ($n = 81$) had $P < 0.05$ in FinnGen with same direction of effect (Fig. 6d and Supplementary Table 9b). Also, 94.81% ($n = 73$) of MILTON 'known binary' associations had supporting evidence ($P < 0.05$ with the same direction of effect). Among the 167 MILTON putative novel associations that could be mapped to FinnGen results, 38.92% ($n = 65$) achieved $P < 0.05$ in FinnGen with the same direction of effect (Fig. 6d and Supplementary Table 9b).

**Putative novel hits rank highly in two orthogonal artificial intelligence-based tools**
We then sought to validate the putative novel hits inferred by PheWASs on MILTON-augmented cohorts, using two independent machine-learning tools, Mantis-ML (v.2.0)[39,40] and AMELIE (Automatic

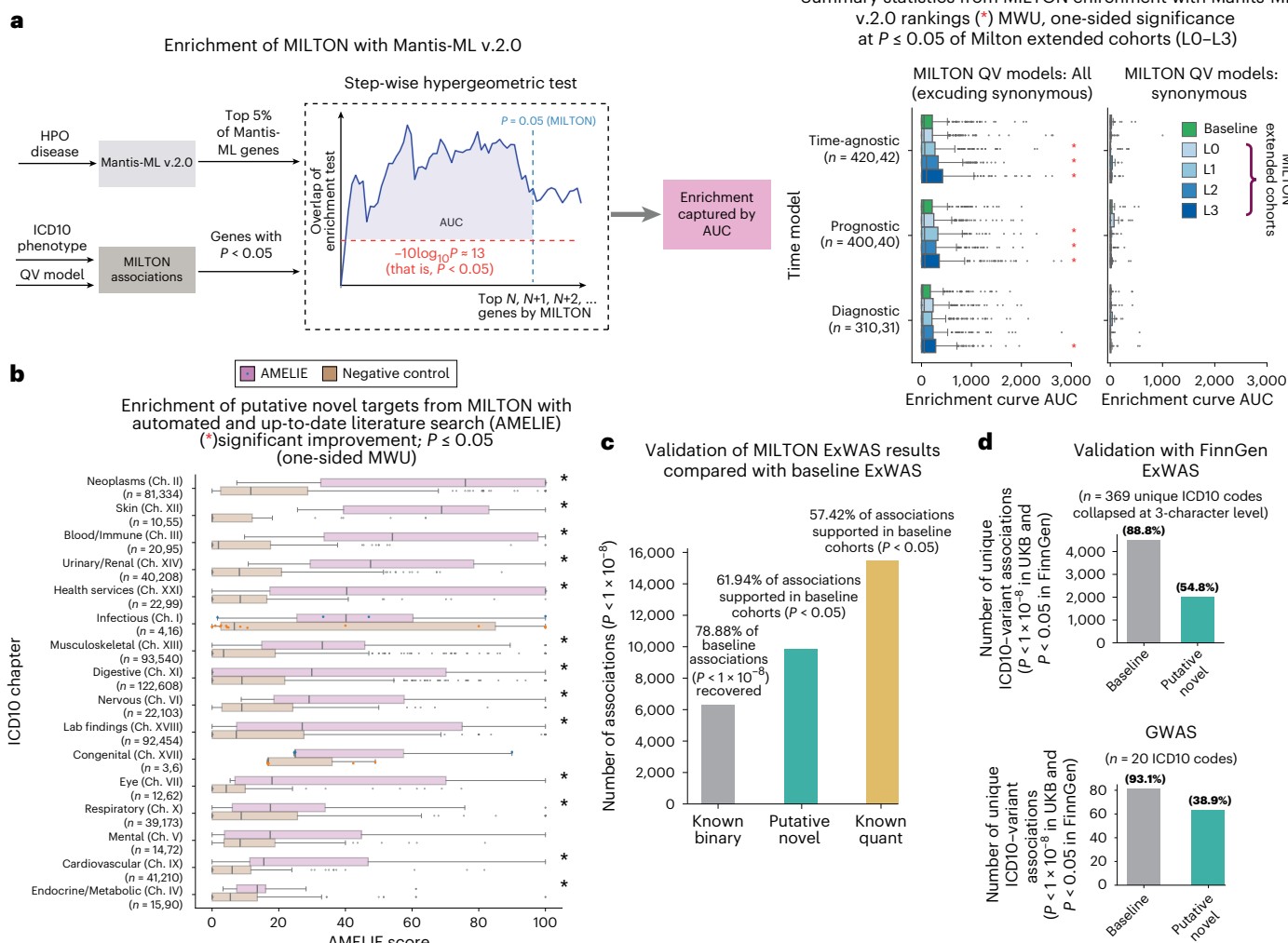

**Fig. 6 | Validation of PheWAS results on MILTON-augmented cohorts using orthogonal machine-learning methods and FinnGen. a**, Left, flowchart depicting the stepwise hypergeometric tests performed to test enrichment of top predictions between MILTON-based PheWAS results (FET, two-sided, unadjusted $P < 0.05$) and top gene–disease associations predicted by Mantis-ML (v.2.0; Methods). Right, box plots comparing the enrichment AUC between MILTON-augmented cohorts and baseline cohorts across all three time-models and across all nonsynonymous QV models or exclusively on the synonymous QV model. Number of samples shown refers to these cases. Comparison is done for 14 HPO terms that could be manually mapped to ICD10 codes. **b**, Breakdown of AMELIE-aggregated scores by ICD10 chapter (sorted by chapter median) for putative novel targets per three-character ICD10 code. Negative controls were generated through ten samplings of random gene sets, equal in size to the respective MILTON gene sets. Points are plotted only for boxes where $n < 10$.

**c**, Validation of MILTON ExWAS results, and putative novel hits compared with baseline ExWAS analysis. $P$ values are from FET (two-sided, unadjusted). **d**, Validation of variant–ICD10 code associations in FinnGen Biobank (release 10 for ExWAS comparison and release 11 for GWAS comparison). FinnGen release 10 $P$ values are from GWAS SAIGE[38] (two-sided, unadjusted) while release 11 $P$ values are from GWAS REGENIE (two-sided, unadjusted). UKB ExWAS $P$ values are from FET (two-sided, unadjusted) and UKB GWAS $P$ values are from REGENIE–Firth (two-sided, unadjusted). Percentages with respect to imputed genotypes and mapped phenotypes with available FinnGen GWAS summary statistics file are given at the top of each bar plot. Box plots show the median as center line and top and bottom quartiles as box limits; whiskers extend to points within 1.5 interquartile ranges of the box limits; points denote outliers. No multiple testing correction was performed.

Mendelian Literature Evaluation)[41,42]. Mantis-ML (v.2.0) is trained phenome-wide, leveraging publicly available gene-centric resources such as the Human Phenotype Ontology (HPO), Open Targets and Genomics England as well as knowledge graphs to assign ranks for each gene in the human exome across thousands of phenotypes. Step-wise hypergeometric tests were performed to compare Mantis-ML (v.2.0)[39,40] predictions and associations identified by MILTON (Fig. 6a, Methods and Supplementary Notes). Notably, genes identified in L3 cohorts for nonsynonymous QV models ($P < 0.05$) were significantly enriched in top-ranking genes from Mantis-ML (v.2.0) compared with the baseline cohort across all three time-models (Fig. 6a; comparison per phenotype shown in Supplementary Fig. 20a). Furthermore, all phenotypes were enriched in top-ranking genes under the ultra-rare

damaging QV model (UKB minor allele frequency ≤ 0.005%; ref. 4 in MILTON-augmented cohorts compared with the synonymous QV model, followed by the ultra-rare variants according to the missense tolerance ratio model (URmtr[4]) and the protein truncating variant model (ptv[4]; Supplementary Fig. 20b,c).

AMELIE (v.3.1.0)[41,42] performs an automated literature search, integrating nightly updates from the entire PubMed corpus, to estimate gene rankings for disease causality among a pool of gene candidates. AMELIE requires diseases to be organized within the HPO. Thus, we first identified the five most semantically similar HPO diseases out of a pool of 17,451 phenotypes to each ICD10 code and queried AMELIE for disease–gene associations for those diseases (Supplementary Notes). We observed that for 13 out of 16 ICD10 chapters, putative novel

targets (Supplementary Table 7a) had significantly higher AMELIE score (*P* < 0.05) than negative controls generated through ten samplings of random gene sets of the same length (Fig. 6b and Supplementary Fig. 21).

These validations pinpoint the power of MILTON to highlight putative novel signals also supported by comprehensive artificial intelligence-based gene-prioritization methods, which leverage large amounts of biological evidence from literature and hundreds of public data resources.

## Discussion

We present here MILTON, a machine-learning framework for multi-omics and biomarker-based disease prediction, and for empowering case–control studies across five UKB ancestries. Despite the feature set employed in this study being fairly broad and diverse and not disease-specific, MILTON achieved relatively high to high predictive power in a considerable number of phenotypes (AUC > 0.7 for 1,091 ICD10 codes, AUC > 0.8 for 384 ICD10 codes and AUC > 0.9 for 121 ICD10 codes). The lack of predictive power in the rest of the phenotypes indicates that not all phenotypes may have a distinctive biomarker fingerprint, at least not in the currently used biomarker set. There are certain disease chapters where models could be improved by inclusion of more informative features across all ancestries and time-models, such as chapter II (neoplasms), chapter VIII (ear) and chapter XII (skin) (Supplementary Fig. 2a and Supplementary Table 2a–e). For example, addition of macromolecular expression data of already known markers of breast cancer (C50)[43] can help improve the model's performance. It is important to note that while MILTON outperformed PRSs for most diseases, for certain diseases, such as melanoma, breast cancer, prostate cancer and primary open-angle glaucoma, PRSs had higher predictive power (AUC improvement ≥ 0.1) than the 67 quantitative traits employed by MILTON. We currently used standard PRSs generated by Thompson et al.[13] instead of enhanced PRSs[13] because the latter scores were available for only 20.75% (*n* = 104,231) of individuals. Perhaps models trained using enhanced PRSs or PRSs derived from other formulations may lead to improved performance, especially for diseases where clear prognostic or diagnostic biomarkers are unavailable. Inclusion of proteomics data also helped to achieve improved performance (ΔAUC ≥ 0.1) for 52 phenotypes. We thus encourage exploring other feature spaces that could further improve performance of our models in additional phenotypes.

On some occasions, applying MILTON reduced the power for signal discovery by diluting gene–disease signals from the baseline analysis which was naive to MILTON. However, overall, 87.41% of the known signals from baseline PheWAS either had an improved signal or retained their genome-wide significance with the same direction of effect. Thus, MILTON more often leads to improved signal detection and discovery yield than to diminished signals, providing strong confidence that this approach is truly augmenting the traditional (baseline) association tests relying solely on positively labeled case definitions in large biobanks. An additional 182 signals were putative novel gene–disease signals that were not discovered by either the baseline case–control or quantitative trait PheWAS. These putative novel associations need to be further experimentally validated.

Recently, advanced deep-learning-based methods for feature imputation have emerged[44], which may help overcome some of the recurrent challenges biobank datasets face around data missingness and imputation. MILTON is distinct from methods that focus on feature imputation[44] of missing binary or quantitative traits (that is, of the independent variables), and instead focuses on augmenting the predicted disease phenotypes (that is, the dependent variable) by leveraging a wealth of input features. Specifically, MILTON first learns a disease signature, given a set of binary and quantitative traits, and then predicts undiagnosed individuals sharing the learnt signature per disease, effectively refining the original case–control cohort definitions. Future improvements of MILTON could include a flexible time-lag,

time-model selection for each ICD10 code to reflect variable dynamics of disease development and the biomarker profiles that characterize these prodromes (Supplementary Notes).

Furthermore, MILTON provides a way to enhance rare-variant collapsing analysis while bridging the gap between clinical biomarkers and diseases. In cases where the top predictive biomarkers currently lack known biological connection with a disease while the predictive model achieves good performance (AUC > 0.8), we suggest these links to be explored further as our knowledge of disease-specific biomarkers continues to grow. Experts are advised to exercise caution while looking at downstream association studies in such cases. The MILTON-inferred biomarker sets may provide insights into defining minimal sets of biomarkers, irrespective of diagnosis, to be collected as part of future biobanks, and to be made available to registered researchers. Finally, MILTON provides the power to characterize disease-specific blood-based feature sets that can define and predict a wide range of human pathologies and provide potential related mechanistic insights. This has implications for future strategies for prevention and early detection of disease.

## Online content

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

¹Centre for Genomics Research, Discovery Sciences, BioPharmaceuticals R&D, AstraZeneca, Cambridge, UK. ²Department of Haematology, University of Cambridge, Cambridge, UK. ³Department of Haematology, Cambridge University Hospitals NHS Foundation Trust, Cambridge, UK. ⁴Clindatapark Ltd, Babraham, Cambridge, UK. ⁵Division of Cardiology, Department of Medicine, Stanford University, Palo Alto, CA, USA. ⁶Clinical Development, Research and Early Development, Respiratory and Immunology (R&I), BioPharmaceuticals R&D, AstraZeneca, Cambridge, UK. ⁷Centre for Genomics Research, Discovery Sciences, BioPharmaceuticals R&D, AstraZeneca, Waltham, MA, USA. ⁸Department of Medicine, Austin Health, University of Melbourne, Melbourne, Victoria, Australia. ⁹These authors contributed equally: Manik Garg, Marcin Karpinski. ✉e-mail: slav.petrovski@astrazeneca.com; dimitrios.vitsios@astrazeneca.com

## Methods

### Ethics

Our research complies with all relevant ethical regulations. Both biobanks have been approved by the relevant board or committee. Ethics approval for the UKB study was obtained from the North West Centre for Research Ethics Committee (11/NW/0382)[45].

The Coordinating Ethics Committee of the Hospital District of Helsinki and Uusimaa approved the FinnGen study protocol (number HUS/990/2017)[38]. The FinnGen study is approved by the Finnish Institute for Health and Welfare (approval numbers THL/2031/6.02.00/2017, amendments THL/1101/5.05.00/2017, THL/341/6.02.00/2018, THL/2222/6.02.00/2018, THL/283/6.02.00/2019 and THL/1721/5.05.00/2019), the Digital and Population Data Service Agency (VRK43431/2017-3, VRK/6909/2018-3 and VRK/4415/2019-3), the Social Insurance Institution (KELA) (KELA 58/522/2017, KELA 131/522/2018, KELA 70/522/2019 and KELA 98/522/2019) and Statistics Finland (TK-53-1041-17).

### UKB cohort

The UKB comprises data from 502,226 participants aged 37–73 years at the time of recruitment with median age being 58 years[46]. Of these, 54.4% are female. The data collected from these participants include, but are not limited to, up-to-date diagnosis information, body size measures, blood count measures, blood biochemistry measures, genomics data[45] as well as proteomics data[14,15] (for 10% of participants). All participants provided informed consent and participation was voluntary.

### FinnGen cohort

FinnGen comprises data from 412,181 individuals (55.9% female) with median age of 63 years[38]. All participants provided informed consent and participation was voluntary. We did not apply for access to patient-level data and used only FinnGen GWAS summary statistics to validate our findings.

### Population descriptors in UKB

We used peddy[47] and 1000 Genomes Project data to classify genetic ancestries at a broad continental level (peddy_prob ≥ 0.9). We additionally removed participant samples where principal component values were more than 4 s.d. from the mean for the first four principal components, for the large EUR ancestry population only.

### Defining cases in UKB

All the participants diagnosed with a given ICD10 code across four different UKB fields (UKB v.673123, released in April 2023) were considered as cases. These UKB fields are 41270 diagnosis: ICD10; 40001 underlying (primary) cause of death: ICD10; 40002 Contributory (secondary) cause of death: ICD10; and 40006 type of cancer: ICD10. Out of 13,942 ICD10 codes, 12,081 codes already had entries for greater than 50% of cases in the field ID 41270 (Supplementary Fig. 23a). When analyzing a parent node such as N18, patients diagnosed with any of its sub-nodes: N18.0, N18.1, N18.2, N18.3, N18.4, N18.5, N18.8, N18.9, were included as well. If, after taking a union, there were at least 100 known cases for each ICD10 code, that ICD10 code was processed further (Supplementary Fig. 24). This resulted in 3,213 unique disease phenotypes analyzed by MILTON across all ancestries (Supplementary Table 1a), 2,392 of which achieved an AUC > 0.60 and are being reported in our public MILTON portal. The following additional filtering steps were applied to these cases when applicable.

### Filtering based on lag between sample collection and diagnosis

To assess the impact of time-lag between biomarker sample collection and diagnosis dates for all ICD10 codes with at least 100 known cases, cases were filtered based on given (time-model, time-lag)

pairs: (1) time-agnostic (no lag): include all cases irrespective of any time difference between biomarker sample collection and diagnosis; (2) prognostic ('$t$' years lag): only include those cases who received diagnosis between 0 years and '$t$' years (inclusive) after biomarker sample collection; (3) diagnostic ('$t$' years lag): only include those cases who received diagnosis between 0 years and '$t$' years (inclusive) before biomarker sample collection.

Therefore, if time-model = prognostic and time-lag = 5 years, then the MILTON pipeline will be trained on all known cases who were diagnosed ≤5 years of biomarker collection (Supplementary Fig. 24). In the case of multiple lag values per subject, the shortest time-lag was used for analysis. This will happen when analyzing a parent node such as 'N18' and the patient has already been diagnosed with its sub-nodes such as N18.1 and N18.3 at two different time-points. As the diagnosis information can be retrieved from any of the four UKB fields (41270, 40001, 40002 and 40006), the corresponding date field was used to retrieve time for diagnosis: date of first in-patient diagnosis (41280), date of death (40000) and date of cancer diagnosis (40005). Please note that the time-lag was calculated as the difference between biomarker collection date (or biomarker sample collection date in case biomarker was measured from blood or urine samples) and diagnosis date. This difference was converted to days and divided by 365, rounding to closest integer, to obtain time-lag in years.

We performed a sensitivity analysis for the effect of the sample collection and diagnosis time-lag on 400 randomly selected ICD10 codes. Within each time-model, no significant difference in performance was observed across time-lags (two-sided MWU: $P > 0.05$; Supplementary Fig. 1a). Furthermore, diagnostic models had consistently higher performance than prognostic models across all time-lags tested (two-sided MWU: $P < 0.05$; Supplementary Fig. 1a). This difference remained consistent across all ICD10 chapters (Supplementary Fig. 1b) and increased with increasing time-lag (Supplementary Fig. 1c). However, for time-lags ≥ 5 years, diagnostic models had significantly lower cases for training than prognostic models (two-sided MWU: $P \leq 1 \times 10^{-3}$; Supplementary Fig. 1d). Additionally, the number of cases available for training increased with increasing time-lag (Supplementary Fig. 1d). This is also evident from the lag distribution across all UKB individuals irrespective of specific diagnosis (Fig. 2a). A longer time-lag resulted in more ICD10 codes satisfying the minimum 100-cases criterion for training MILTON models (Methods). Therefore, we selected 10 years as the optimal time-lag, since: (1) median performance of prognostic models started dropping after the 10-year time-lag; (2) it had comparable performance to other time-lags for diagnostic models; and (3) the number of cases attained with a 10-year time-lag was significantly higher compared with the 1- or 5-year time-lags (Supplementary Fig. 1d).

### Filtering based on sex-specificity of an ICD10 code

Some diseases might affect the opposite sex on rare occasions or not at all, given anatomical differences between men and women. To have sufficiently clean data for training a machine-learning model, cases from the opposite sex were filtered out if an ICD10 code was deemed to dominantly affect one sex (Supplementary Fig. 24). To find a suitable threshold for filtering, 117 male-specific and 606 female-specific ICD10 codes were shortlisted by keyword search. For male-specific diseases, 'male', 'prostate', 'testi' and 'patern' keywords were used. For female-specific diseases, keywords 'female', 'breast', 'ovar', 'uter', 'pregnan' and 'matern' were used. Only those ICD10 codes with at least 100 cases in total were considered for this analysis, resulting in 31 male-specific and 201 female-specific ICD10 codes (Supplementary Table 10). As shown in Supplementary Fig. 23b, 231 out of 232 diseases had proportion of dominant sex above 0.9, with highest number of women = 12 for N40 hyperplasia of prostate ($n$ male = 26,033) and highest number of men = 128 for C50.9 breast, unspecified ($n$ female = 15,311). The ICD10 code N62 hypertrophy of breast ($n$ male = 317, $n$ female = 864) did not satisfy the ±0.9 threshold and includes a condition called gynecomastia, which

causes abnormal enlargement of breasts in men. Therefore, N62 was not considered to be a female-specific phenotype in our analysis.

## Defining controls in UKB

All the remaining UKB participants who were not diagnosed with any of the ICD10 codes in the entire chapter that the current ICD10 code belongs to were considered as 'potential controls' (Supplementary Fig. 24). To maintain similar case to control ratio across different ICD10 codes, the number of controls was restricted to be a maximum of control-case ratio (ctrl_ratio) times the number of cases, where ctrl_ratio$_{max}$ = 19 for robust results when sufficient data were available (mean Pearson's $r$ between each replicate pair > 0.9; Supplementary Fig. 25a,b) or ctrl_ratio$_{max}$ = 9 otherwise (Supplementary Fig. 25c). These 'controls' were randomly selected from the list of 'potential controls' and further filtering was applied when applicable. The final ctrl_ratio may be less than ctrl_ratio$_{max}$ depending on how many controls remained after applying all filters.

## Filtering to match baseline characteristics of cases

According to the UKB field ID 31: Sex, there are 54.40% female ($n$ = 273,326) and 45.60% male ($n$ = 229,086) across the entire UKB cohorts. This imbalance was significant ($P < 1 \times 10^{-3}$) across multiple age groups (Supplementary Fig. 23c) as well as multiple ICD10 codes with varying case numbers (Supplementary Fig. 23d). To maintain similar distributions of age (UKB field 21003) and sex (UKB field 31) across cases and controls, age was divided into four bins. Across these four bins, the distribution of women (and men) in controls was matched with the distribution of women (and men) in cases. For example, if ten women and five men were diagnosed with a given ICD10 code and their hypothetical distribution across four age bins is given in Supplementary Table 11, then the controls were sampled to match the case distribution as given in Supplementary Table 11. This criterion meant that sometimes cases were dropped to get the matching distribution between cases and controls (Supplementary Fig. 24).

## Features

**UKB 67 traits.** For each participant, quantitative traits ($n$ = 67) collected from blood assays ($n$ = 50), urine assays ($n$ = 4) and physical measures ($n$ = 10) along with covariates fasting time, sex and age were used as features for training the machine-learning models (Supplementary Table 12).

 If a feature was collected during multiple assessment visits (instances 0–3 in UKB), only the first non-null values for each trait were used. The correlation between these features is shown in Supplementary Fig. 26. Low-density lipoprotein direct was highly correlated with apolipoprotein B (Pearson's correlation coefficient = 0.96) and cholesterol (Pearson's correlation coefficient = 0.95). Similarly, HDL cholesterol was highly correlated with apolipoprotein A (Pearson's correlation coefficient = 0.92) and negatively correlated with triglycerides (Pearson's correlation coefficient = −0.44) and waist circumference (Pearson's correlation coefficient = −0.48).

 Despite taking the first non-null value across multiple instances, rheumatoid factor and estradiol had more than 80% values missing (Supplementary Fig. 22). It has been recommended by the UKB to treat these missing values as 'naturally low' instead of missing[48]. Furthermore, we observed that the amount of missingness correlated among features obtained from the same assay category, suggesting that these participants did not have the corresponding tests taken (Supplementary Fig. 27). It is unclear why microalbumin in urine had 68.84% missing values while other features obtained from urine assays only had around 3.50% missing values. Its missingness pattern was also similar to those of estradiol and rheumatoid factor.

 It is important to note that UKB was designed for research purposes, and the same measurements were collected during baseline visits of each patient to one of the available recruitment centers,

regardless of any underlying diagnoses/conditions. These measurements have been collected uniformly (except for some missing data for a small number of patients) and in a standardized manner by UKB for all patients. No additional measurements from any other labs, for example, from in-patient hospitalization visits, have been included. Thus, there is no mixture of research-based measurements with uncommon clinical measurements that would require further harmonization.

**UKB plasma proteomics data.** Normalized protein expression values for 2,923 proteins were available for 46,327 UKB individuals of EUR ancestry[14,15]. If a single protein was recorded across multiple panels, the column with the lowest proportion of missing values was used. Age, sex, body mass index, smoking status (20116), alcohol intake (1558), paternal history (20107), maternal history (20110) and blood type (23165) were added as covariates.

**UKB date fields for calculating time-lags.** Blood sample collection dates and urine sample collection dates were recorded in UKB fields 21842 and 21851, respectively. The dates for recording physical measures, fasting time, sex and age were assumed to be the date when a participant visited the assessment center (UKB field 53). As the difference between these dates is 0 for most of the participant IDs (Supplementary Table 13), blood sample collection dates were used for all features when calculating time-lag between sample collection date and diagnosis date (Methods section 'Filtering based on lag between sample collection and diagnosis').

## Feature pre-processing

UKB has taken extensive measures to ensure quality of values (https://biobank.ctsu.ox.ac.uk/crystal/crystal/docs/biomarker_issues.pdf). To account for any missing values and transform the data for machine-learning pipelines, the following pre-processing steps were applied on the training data and then on the testing data: (1) Missing values in each feature (except rheumatoid factor, estradiol and testosterone) were imputed with median value of that feature (Supplementary Table 12). As it is recommended to treat missing values in rheumatoid factor and estradiol as 'naturally low' instead of missing[48]: rheumatoid factor was imputed with 0 IU ml$^{-1}$ (https://www.ouh.nhs.uk/immunology/diagnostic-tests/tests-catalogue/rheumatoid-factor.aspx) and estradiol was imputed with values 36.71 pmol l$^{-1}$ for men and 110.13 pmol l$^{-1}$ for women (https://www.southtees.nhs.uk/services/pathology/tests/oestradiol/). Also, the testosterone feature was imputed with respective median values for men and women. (2) Categorical features were one-hot encoded. (3) All features were standardized to have zero mean and unit variance.

## Model training

In MILTON, we first select cases and controls for each ICD10 code[10] (Data-Coding 19), and then extract biomarker values for each participant to derive a disease-specific signature. Next, we use an eXtreme Gradient Boosting (XGBoost[49]) model on a single balanced case–control set to find the optimal hyperparameters for a given ICD10 code. We train an ensemble model with fivefold cross-validation using up to four balanced case–control sets (bound defined by the initial number of cases and total UKB cohort size) and repeat for ten stochastic iterations to ensure the entire control set is fed to the model. We then apply the final model to the entire UKB cohort to predict the probability of each UKB participant being a case for that given disease. We then assign individuals as 'cases' based on four different probability thresholds (named L0–L3). The L0 class includes the strictest cut-off whereas L3 uses the most lenient, including the largest number of predicted cases. Finally, we repeat rare-variant collapsing analysis[4] on each of the augmented cohorts and compare the results against the baseline cohorts used to train the models, as well as against the original PheWAS results that relied exclusively on positive labeled cases[4].

Further, the training of the XGBoost classifier was divided into either: (1) two steps when training on 67 traits: hyperparameter tuning and model prediction; or (2) three steps when training on UKB protein expression data: feature selection, hyperparameter tuning and model prediction.

## Package requirements

For development of the MILTON pipeline, the following python packages were used: python (v.3.10.13), pandas (v.2.1.4), numpy (v.1.22.4), scipy (v.1.11.4), scikit-learn (v.1.3.0), scikit-image (v.0.22.0), xgboost (v.1.7.3), pyarrow (v.14.0.2), holoviews (v.1.18.3), matplotlib (v.3.8.0), pytest (v.7.4.0), Jinja2 (v.3.1.3), pyparsing (v.3.0.9), dask (v.2023.11.0), dask-ml (v.2023.3.24), dask-jobqueue (v.0.8.1), numba (v.0.56.0), tqdm (v.4.65.0), beautifulsoup4 (v.4.12.2), boruta_py (v.0.3).

For data analysis and visualization, the following python packages were used: python (v.3.10.13), pandas (v.2.1.4), numpy (v.1.22.4), matplotlib (v.3.8.0), seaborn (v.0.12.2), statannotations (v.0.5.0), upSetPlot (v.0.8.0), missingno (v.0.5.1), scipy (v.1.11.4).

## Feature selection (for proteomics data only)

Training cases were divided into 50% for feature selection and 50% for hyperparameter tuning and model prediction. The Boruta feature selection algorithm with random forest classifier was trained on this subset of training cases and an equal number of randomly sampled controls (Methods section 'Defining controls in UKB'). This process was repeated nine times and a set union of 'confirmed' features from all iterations were used for further analysis.

## Hyperparameter tuning

The number of XGBoost trees ('n_estimators') was tuned for each ICD10 code. For this, equal numbers of age–gender matched controls as cases were shortlisted for each ICD10 code. The XGBoost classifier was then fit on the entire dataset using each of the n_estimators values {50, 100, 200, 300} and its performance was recorded in terms of area under the receiver operating characteristic curve (AUROC). The n_estimators value giving the highest AUROC for each ICD10 code was then used for further analysis.

## Model prediction

Once the optimal number of n_estimators was determined for each ICD10 code, the next step was to predict the probability for each UKB participant of being a case given their biomarker profile. A predicted probability value closer to 1 denotes higher chances of a participant being a case while a value closer to 0 denotes higher chances of being a control. To obtain these predicted probabilities, steps outlined in the paragraph below were repeated ten times and average prediction probability across these ten iterations was calculated for each participant in the entire UKB.

In each iteration, all the cases ($n$) diagnosed with that ICD10 code were taken (Methods section 'Defining cases in UKB') and controls (ctrl_ratio × $n$) were randomly sampled (Methods section 'Defining controls in UKB'). Fivefold cross-validation was performed, where in each fold, four-fifths of the data (cases = $n$ × 4/5, controls = ctrl_ratio × $n$ × 4/5) were used for training a 'Balanced Ensemble Classifier' and the remaining one-fifth (cases = $n$ × 1/5, controls = ctrl_ratio × $n$ × 1/5) were used for testing model performance. A Balanced Ensemble Classifier trains an XGBoost classifier on an equal number of cases ($n$ × 4/5) and controls ($n$ × 4/5) by random subsampling from ctrl_ratio × $n$ × 4/5 controls without replacement and repeats the training process ctrl_ratio times to include all controls. The average performance from these four Balanced Ensemble Classifier repeats is reported as the performance of each cross-validation fold and the average performance across all five folds is reported as the performance of each overall iteration. An XGBoost classifier is then fitted on the entire training dataset (cases = $n$, controls = ctrl_ratio × $n$) to generate prediction probabilities for all UKB participants.

We also compared the predictive performance of XGBoost with other classification models, such as logistic regression (LR) and the autoencoder DL model. We observed that XGBoost gave a better performance than LR and comparable performance to DL ($AUC_{XGBoost}$ versus $AUC_{LR}$: $0.655 \pm 0.09$ versus $0.646 \pm 0.09$, MWU two-sided $P = 0.000012$; $AUC_{XGBoost}$ versus $AUC_{DL}$: $0.655 \pm 0.09$ versus $0.656 \pm 0.08$, MWU two-sided $P = 0.025$; Supplementary Fig. 28).

Additionally, we compared performance reported on cross-validation set (averaged across five folds before calculating median across ten replicates) with performance reported on a held-out test set when only 85% of data were used for training MILTON models and the remaining 15% of unseen data were used for testing model performance. Across 100 ICD10 codes, we observed performance measures to be comparable (mostly within ±0.1 difference range; Supplementary Fig. 29) across the two sets and decided to use all available data for training MILTON while reporting performance measures from cross-validation sets.

## MILTON-augmented case cohort generation

For each ICD10 code, four novel case ratio (NCR)-based cohorts, namely 'L0', 'L1', 'L2' and 'L3', were generated. Here, NCR is defined as 1 + ((number of new cases)/(number of known cases)). The distribution of sex for each new cohort was matched with that of known cases.

To generate NCR-based MILTON cohorts with known and new cases for each ICD10 code, all known cases were assigned an average prediction probability of 1 and any not-known participant with probability less than 0.7 was filtered out. All the remaining not-known participants were ranked in decreasing order of their average prediction probability, that is, a participant with highest prediction probability will have rank 1, second highest prediction probability will have rank 2 and so on. These ranks were then divided by the number of known cases to obtain an (NCR − 1) score per participant. Therefore, if the total number of known cases is ten, then adding all not-known participants with rank less than or equal to 3 will give an NCR of 1 + (3/10) = 1.3. The range of NCR was clipped between 1 (no new cases) and 10 (number of new cases = 9 × number of known cases), and it was divided into four quantiles: $NCR_{q1}$, $NCR_{q2}$, $NCR_{q3}$, $NCR_{q4}$. All not-known cases with rank less than equal to ($NCR_{q1}$ − 1) × (number of known cases) were added to 'L0' cohort, all not-known cases with rank less than equal to ($NCR_{q2}$ − 1) × (number of known cases) were added to 'L1' cohort and so on for the 'L2' and 'L3' cohorts. All known cases were then added to each of these NCR-based case cohorts.

To assess the suitability of using dichotomized MILTON-augmented cohorts, that is, considering an individual as a case or control, we also repeated rare-variant collapsing analysis by taking the MILTON-predicted probability scores as continuous measures (Supplementary Notes). Upon comparing gene–ICD10 associations ($P < 10^{-8}$) obtained from this new approach with those obtained using dichotomized cohorts, only 30.37% ($n = 82$) of baseline associations were recovered in this approach having the same direction of effect, compared with 87.41% ($n = 236$) recovered when using the dichotomized cohorts (Supplementary Fig. 10 and Supplementary Table 14). Furthermore, we noticed that while we recovered 99.62% ($n = 1,043$) of known quantitative and 73.63% ($n = 134$) of putative novel associations with the new approach, it also led to a staggeringly high number of putative novel associations ($n = 7,365$). Given the low sensitivity achieved when using prediction scores as a continuous phenotype, we expect a very large number of false positives among the putative novel hits. We believe that these signals may be contributed by the genetic architecture of individuals with lower probability scores and, therefore, decided to proceed with the original dichotomized cohorts.

## PheWAS collapsing analysis on case–control cohorts

The PheWAS collapsing analysis was performed on each {ICD10 code, case cohort, QV model} combination using whole-genome sequencing

data from UKB participants of different ancestries (EUR = 419,391; SAS = 8,577; AFR = 7,731; EAS = 2,312; AMR = 706). All noncases according to each {ICD10 code, case cohort} pair were taken as controls and matched with sex distribution of cases. A 2 × 2 contingency table was generated between cases and controls with or without QVs in each gene[4]. Fisher's exact test (two-sided) was performed with the null hypothesis being that there is no difference in the number of QVs in each gene between cases and controls[4]. All genes with a $P$ value less than $1 × 10^{-8}$ were considered to be significantly associated with the given ICD10 code for a given QV model and case–control cohort.

Clonal hematopoiesis is the presence of clonal somatic mutations in the blood of individuals without a hematologic malignancy[50], and these somatic variants can be detected with germline variant callers[51]. To avoid conflating somatic with germline variants, and age-confounded disease associations, we removed from further analysis 16 genes we previously identified in the UKB as carrying clonal somatic mutations[15,52] (*ASXL1*, *BRAF*, *DNMT3A*, *GNB1*, *IDH2*, *JAK2*, *KRAS*, *MPL*, *NRAS*, *PPM1D*, *PRPF8*, *SF3B1*, *SRSF2*, *TET2*, *TP53* and *CALR*). A few other problematic genes as listed in ref. 4 have also been removed from further investigation.

### GWAS analysis on case–control cohorts
We selected 20 binary phenotypes for common variant GWAS analysis by randomly selecting one ICD10 code from each of the four AUC-specific and five training case-specific bins obtained from time-agnostic, EUR ancestry models (Supplementary Table 15). We used UKB imputed genotypes (UKB Field 22828) to perform GWAS across baseline and L0–L3 MILTON cohorts using REGENIE (v.3.1)[53] with sex, age, age × sex, age², age² × sex, array batch and ancestry principal components (principal components 1–10, as supplied by the UKB). For step 1, we used high-quality genotyped (UKB field 22000) variants with minor allele frequency (MAF) > 0.01, minor allele count (MAC) > 100, missing genotype fail rate < 1% and $P$ Hardy–Weinberg > $1 × 10^{-15}$, which we pruned based on linkage disequilibrium (LD) using PLINK2 ($R^2$ < 0.8, using a 1-Mb window with 100-kb step size). For step 2, we included all imputed single nucleotide polymorphisms with an INFO score > 0.7 and a minimum MAC of 400 with the same covariates as for step 1 and used approximate Firth regression to adjust for case–control imbalance.

To define association loci for each trait, we first selected all variants with $P$ < $5 × 10^{-8}$ across 'known' and L0–L3 cohorts. We then iteratively clumped these variants based on the 2-Mb region regardless of the cohort defining the variant with the smallest $P$ value to index the locus across all cohorts. We annotated the closest gene using Ensembl (GRCh37) Biomart using the biomaRt package.

### FIS calculation
During model training, the XGBoost classifier was enabled to return the relative biomarker (feature) importance scores (FISs) for each feature. FIS was calculated for each 'n_estimator' in XGBoost using the Gain measure, which quantifies average performance improvement when a given feature is used for making a decision split. This process was repeated for all 'n_estimators' and average FISs were reported for each trained XGBoost model. Because we used a balanced Ensemble classifier to train 19 XGBoost models on 19 different subsamples of controls (ctrl_ratio parameter; Methods section 'Model prediction') within fivefold cross-validation for each of ten replicates, the mean of those 19 × 5 × 10 average FISs was reported for each ICD10 code and feature as the final importance score.

### Finding optimal FIS cut-off
For each time-model, nonzero FISs per ICD10 code were first log-transformed and then standardized to have zero mean and unit standard deviation. Transformed FISs for top features per ICD10 code per time-model were extracted and compared with the median AUC of ICD10 codes across ten replicates. It was observed that transformed

FISs of top features highly correlated with AUC performance of ICD10 codes (Pearson's correlation coefficient > 0.80; Supplementary Fig. 7e). This suggested that high-performing models with AUC > 0.80 identified dominating features and assigned high FISs to them (>4 s.d. away from the mean FIS). A threshold of 1.2 was therefore chosen for further analysis because it captured dominating features for most ICD10 codes with median AUC > 0.6 for which we made results available. This value comes from rounding up to the first decimal place the 25th percentile of feature importance $Z$-scores for top features of ICD10 codes with AUC 0.60–0.65 (Supplementary Fig. 7e). By selecting this cut-off, we confirmed as a positive control for *PKD1* (which had a relatively large number of putative novel associations in the unfiltered set of results) that the number of putative novel associations remained below the number of known binary associations, while it started exceeding them for higher FIS $Z$-score cut-offs (Supplementary Fig. 7f).

### Comparison with Mantis-ML (v.2.0)
Stepwise hypergeometric tests proceed through identifying a ranked list of genes (ordered from most to least associated according to Mantis-ML (v.2.0)[39,40]) and a set of genes 'known' to be associated. The latter may be a list of tentatively associated genes, provided, for example, by a PheWAS with a relaxed $P$ value threshold. The test then iteratively goes through the ranked list, marking the $N$, $N + 1$, $N + 2$ and so on genes as associated in the ranked list, and quantifies the overlap between this and the latter (the set of known associated genes) using Fisher's exact test. In this way, we may observe whether there is substantial overlap, not by chance, between the two without specifying a threshold for the ranked list. The output of this exercise is a sequence of $P$ values that can be converted to scores using $-10\log_{10}P$ transformation. Finally, to provide a simple summary statistic for the test we may take the AUC, where the curve is defined as max($-10\log_{10}(0.05)$, $-10\log_{10}P$), that is, the area exceeding $\alpha$ = 0.05 significance. Requiring that the curves exceed 0.05 significance allows for a certain degree of noise to be filtered out. In this setting, genes from MILTON with a weak significance were used and the top 5% of Mantis-ML (v.2.0)-associated genes were used as the ranked list. Of note, Mantis-ML (v.2.0) uses HPO while MILTON uses ICD10 codes, and thus manual HPO-to-ICD10 mapping was first performed (Supplementary Table 16).

### Comorbidity enrichment analysis
The comorbidity enrichment analysis performs a Fisher's exact test for all distinct ICD10 diagnoses within the selected cohort of UKB participants to find diseases that have significantly higher incidence in the cohort as compared with the general population (the rest of the UKB participants). ICD10 codes are scored according to the results of their Fisher's tests to highlight the most relevant ones. Fisher's exact test produces by default two numbers: $P$ value and odds ratio. The former is a measure of how significant the disease enrichment in the selected cohort is, while the latter corresponds to the effect size, that is, how much more likely the participants from the selected cohort are to be diagnosed with the disease. The score then is taken to be the log of odds ratio, capped at a high value, and normalized to 100 to facilitate visualization of results.

### Statistics and reproducibility
Data distribution was assumed to be normal, but this was not formally tested. $N$ = 484,230 whole-genome sequencing samples from the UKB were derived after quality control checks and filtering for relatedness.

Those ICD10 codes that had less than 100 diagnosed cases were excluded from analysis (Methods section 'Defining cases in UKB'). Under prognostic or diagnostic time-models, those individuals who were diagnosed after or before sample collection were excluded, respectively (Methods section 'Filtering based on lag between sample collection and diagnosis'). For ICD10 codes inferred to be sex-specific, any individuals from the opposite sex were excluded from analysis

(Methods section 'Filtering based on sex-specificity of an ICD10 code'). For a given ICD10 code, those individuals who were diagnosed with any code within the same ICD10 chapter were excluded from the control set (Methods section 'Defining controls in UKB').

Randomization was performed ten times to select case-matched controls for each ICD10 code (Supplementary Fig. 24 and Methods section 'Model prediction').

The investigators were not blinded to allocation during experiments and outcome assessment. Data collection and analysis were not performed blind to the conditions of the experiments.

### Reporting summary

Further information on research design is available in the Nature Portfolio Reporting Summary linked to this article.

## Data availability

All the biomarker information, diagnosis information along with relevant dates and whole-genome sequencing data can be obtained from the UKB (http://www.ukbiobank.ac.uk/register-apply). The list of 67 quantitative traits along with their UKB field IDs is given in Supplementary Table 10 and can be found on the UKB showcase portal (https://biobank.ndph.ox.ac.uk/showcase/search.cgi). UKB plasma proteomics data can also be found here: https://biobank.ndph.ox.ac.uk/showcase/label.cgi?id=1838. Data for this study were obtained under Resource Application Numbers 68601 and 26041. FinnGen GWAS summary statistics results can be downloaded from here: https://www.finngen.fi/en/access_results. Baseline quantitative PheWAS results can be accessed through the AZ PheWAS portal: https://www.azphewas.com. Ensembl Human GRCh37: https://grch37.ensembl.org/Homo_sapiens/Info/Index. All results produced in this study are available in the supplementary tables and can be visualized on the MILTON web-portal (http://milton.public.cgr.astrazeneca.com). PheWAS/ExWAS (allelic model) results for each gene/variant as well as FISs for each ICD10 code can also be downloaded from the MILTON public portal. To aid with visualization, PheWAS/ExWAS results are shown for 67 biomarkers while FISs are shown for 67 biomarkers with or without UKB protein expression data. All results on the portal are for EUR ancestry only, which composed the majority of results (see the supplementary tables for all ancestries). Source data corresponding to all main figures are provided with this paper.

## Code availability

The MILTON source code and example data are publicly available at Zenodo[54] (https://doi.org/10.5281/zenodo.13134143) and GitHub (https://github.com/astrazeneca-cgr-publications/milton-release), under the Mozilla Public License v.2.0. MILTON accesses UKB individual-level data; therefore, users first need to apply to UKB to request access to the UKB input files. Detailed documentation is included in our MILTON repository to convert those files into an appropriate format, ready to be used by the MILTON package. Mock data files with the expected format are already provided in our repository for reference and testing. All results are also publicly available at http://milton.public.cgr.astrazeneca.com.

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

## Acknowledgements

We thank the UK Biobank team and participants for making this valuable resource possible. We thank the participants and investigators of the FinnGen study. We also thank the AstraZeneca Centre for Genomics Research Analytics and Informatics teams for their feedback and help with accessing certain datasets. The authors received no specific funding for this work.

## Author contributions

D.V. and S.P. conceptualized and designed the study. M.K., M.G. and D.V. developed the method and generated phenome-wide results. M.G., M.K., D.M., L.M., O.S.B., F.H., E.W., K.R.S., A.O'N., Q.W., A.R.H., R.S.D., S.P. and D.V. contributed to the methods and analytical strategies. D.M., M.K. and D.V. developed the web-portal. L.M., M.G., M.K. and O.S.B. performed validation analyses. M.G., M.K., L.M., O.S.B., J.M., A.R.H., R.S.D., S.P. and D.V. contributed to exploration and interpretation of results. M.G., L.M., J.M., M.A.F., E.A.A., A.R.H., R.S.D., S.P. and D.V. wrote the paper. All authors provided input and revisions for the final paper.

## Competing interests

M.G., M.K., D.M., L.M., O.S.B., F.H., E.W., K.R.S., M.A.F., J.M., A.O'N., E.A.A., A.R.H., Q.W., R.S.D., S.P. and D.V. are current employees and/or stockholders of AstraZeneca. E.A.A. is a founder of Personalis, Inc., DeepCell, Inc. and Svexa Inc.; a founding advisor of Nuevocor; a nonexecutive director at AstraZeneca; and an advisor to SequenceBio, Novartis, Medical Excellence Capital, Foresite Capital and Third Rock Ventures.

## Additional information

**Correspondence and requests for materials** should be addressed to Slavé Petrovski or Dimitrios Vitsios.

**Peer review information** Peer reviewer reports are available.

# Reporting Summary

## Statistics

For all statistical analyses, confirm that the following items are present in the figure legend, table legend, main text, or Methods section.

| n/a | Confirmed | |
|---|---|---|
| ☐ | ☒ | The exact sample size (*n*) for each experimental group/condition, given as a discrete number and unit of measurement |
| ☐ | ☒ | A statement on whether measurements were taken from distinct samples or whether the same sample was measured repeatedly |
| ☐ | ☒ | The statistical test(s) used AND whether they are one- or two-sided *Only common tests should be described solely by name; describe more complex techniques in the Methods section.* |
| ☐ | ☒ | A description of all covariates tested |
| ☐ | ☒ | A description of any assumptions or corrections, such as tests of normality and adjustment for multiple comparisons |
| ☐ | ☒ | A full description of the statistical parameters including central tendency (e.g. means) or other basic estimates (e.g. regression coefficient) AND variation (e.g. standard deviation) or associated estimates of uncertainty (e.g. confidence intervals) |
| ☐ | ☒ | For null hypothesis testing, the test statistic (e.g. *F*, *t*, *r*) with confidence intervals, effect sizes, degrees of freedom and *P* value noted *Give P values as exact values whenever suitable.* |
| ☒ | ☐ | For Bayesian analysis, information on the choice of priors and Markov chain Monte Carlo settings |
| ☒ | ☐ | For hierarchical and complex designs, identification of the appropriate level for tests and full reporting of outcomes |
| ☐ | ☒ | Estimates of effect sizes (e.g. Cohen's *d*, Pearson's *r*), indicating how they were calculated |

*Our web collection on statistics for biologists contains articles on many of the points above.*

## Software and code

Policy information about availability of computer code

| | |
|---|---|
| Data collection | MILTON accesses UKB individual level data, therefore, users first need to apply to UK Biobank to request for access to the UKB input files. Detailed documentation is included in our MILTON public code repository to convert those files into an appropriate format, ready to be used by the MILTON package. Mock data files with the expected format are already provided in our repository for reference and testing. All generated results are available in our public results portal: http://milton.public.cgr.astrazeneca.com |
| Data analysis | The MILTON method is publicly available at https://zenodo.org/records/13134144 and https://github.com/astrazeneca-cgr-publications/milton-release, under the Mozilla Public License 2.0. |
| | For data analysis and visualization, the following python packages were used: python (v3.10.13), pandas (v2.1.4), numpy (v1.22.4), matplotlib (v3.8.0), seaborn (v0.12.2), statannotations (v0.5.0), upSetPlot (v0.8.0), missingno (v0.5.1), scipy (v1.11.4). For GWAS analysis, REGENIE v3.1 was used and for quantitative PheWAS PEACOK v2.0.0 was used. |

For manuscripts utilizing custom algorithms or software that are central to the research but not yet described in published literature, software must be made available to editors and reviewers. We strongly encourage code deposition in a community repository (e.g. GitHub). See the Nature Portfolio guidelines for submitting code & software for further information.

## Data

Policy information about availability of data

All manuscripts must include a data availability statement. This statement should provide the following information, where applicable:
- Accession codes, unique identifiers, or web links for publicly available datasets
- A description of any restrictions on data availability
- For clinical datasets or third party data, please ensure that the statement adheres to our policy

All the biomarker information, diagnosis information along with relevant dates and whole genome sequencing data can be obtained from the UKB (http://www.ukbiobank.ac.uk/register-apply). The list of 67 quantitative traits along with their UKB field ids is given in Supplementary Table 10 and can be found on the UKB showcase portal (https://biobank.ndph.ox.ac.uk/showcase/search.cgi). UKB plasma proteomics data can also be found here: https://biobank.ndph.ox.ac.uk/showcase/label.cgi?id=1838. Data for this study were obtained under Resource Application Number 26041.

FinnGen GWAS summary statistics results can be downloaded from here: https://www.finngen.fi/en/access_results.

Baseline quantitative PheWAS results can be accessed through AZ PheWAS portal: https://www.azphewas.com.

Ensembl Human GRCh37: https://grch37.ensembl.org/Homo_sapiens/Info/Index.

All results produced in this study are available in Supplementary Tables and can be visualized on the MILTON web-portal (http://milton.public.cgr.astrazeneca.com). PheWAS/ExWAS (allelic model) results for each gene/variant as well as feature importance scores for each ICD10-code can also be downloaded by clicking on the downward arrow on their corresponding web-pagefrom the MILTON public portal. To aid with visualization, PheWAS/ExWAS results are shown for 67 biomarkers while feature importance scores are shown for 67 biomarkers with or without UKB protein expression data. All results on the portal are for European ancestry only, thatwhich comprised the majority of results (see Supplementary Tables for all ancestries). Source data corresponding to all main figures has also been provided.

## Research involving human participants, their data, or biological material

Policy information about studies with human participants or human data. See also policy information about sex, gender (identity/presentation), and sexual orientation and race, ethnicity and racism.

| | |
|---|---|
| Reporting on sex and gender | Related policy is specified in the UK Biobank project (http://www.ukbiobank.ac.uk) |
| Reporting on race, ethnicity, or other socially relevant groupings | Related policy is specified in the UK Biobank project (http://www.ukbiobank.ac.uk) |
| Population characteristics | Related policy is specified in the UK Biobank project (http://www.ukbiobank.ac.uk). |
| | UKB cohort: The UKB comprises of data from 502,226 participants aged 37-73 years at the time of recruitment with median age being 58 years. Of these, 54.4% are females. The data collected from these participants includes, but not limited to, up-to-date diagnosis information, body size measures, blood count measures, blood biochemistry measures, genomics data as well as proteomics data (for 10% of participants). All participants provided informed consent and participation was voluntary. |
| | FinnGen cohort: The FinnGen comprises of data from 412,181 individuals (55.9% females) with median age of 63 years. All participants provided informed consent and participation was voluntary. We did not apply for access to patient-level data and only used FinnGen GWAS summary statistics to validate our findings. |
| Recruitment | Related policy is specified in the UK Biobank project (http://www.ukbiobank.ac.uk) |
| Ethics oversight | Related policy is specified in the UK Biobank project (http://www.ukbiobank.ac.uk) |

Note that full information on the approval of the study protocol must also be provided in the manuscript.

# Field-specific reporting

Please select the one below that is the best fit for your research. If you are not sure, read the appropriate sections before making your selection.

☒ Life sciences          ☐ Behavioural & social sciences          ☐ Ecological, evolutionary & environmental sciences

For a reference copy of the document with all sections, see nature.com/documents/nr-reporting-summary-flat.pdf

# Life sciences study design

All studies must disclose on these points even when the disclosure is negative.

| | |
|---|---|
| Sample size | N=484,230 whole genome sequencing samples from UK Biobank, derived after QC checks and filtering for relatedness. |

| | |
|---|---|
| Data exclusions | We applied our method to high quality, predominantly unrelated gexome sequencing samples from 5 different ancestries with linked health record data in UKB. |
| Replication | We developed MILTON as an ensemble of ensemble classifiers, looking at multiple random sub-samples of the entire cohort during training, applying k-fold cross-validation, and repeating the process for 10 stochastic iterations, to increase robustness of the prediction results. |
| Randomization | Models are trained on all cases during each of the 10 iterations. However in each iteration, controls are randomly selected and matched to cases by size (nine times or nineteen times the number of cases), age and sex. The whole training process is repeated for 10 stochastic iterations to allow inclusion of different and diverse parts of the entire cohort during learning. |
| Blinding | Data collection and analysis were not performed blind to the conditions of the experiments. |

# Reporting for specific materials, systems and methods

We require information from authors about some types of materials, experimental systems and methods used in many studies. Here, indicate whether each material, system or method listed is relevant to your study. If you are not sure if a list item applies to your research, read the appropriate section before selecting a response.

## Materials & experimental systems

| n/a | Involved in the study |
|---|---|
| ☒ ☐ | Antibodies |
| ☒ ☐ | Eukaryotic cell lines |
| ☒ ☐ | Palaeontology and archaeology |
| ☒ ☐ | Animals and other organisms |
| ☒ ☐ | Clinical data |
| ☒ ☐ | Dual use research of concern |
| ☒ ☐ | Plants |

## Methods

| n/a | Involved in the study |
|---|---|
| ☒ ☐ | ChIP-seq |
| ☒ ☐ | Flow cytometry |
| ☒ ☐ | MRI-based neuroimaging |

## Plants

| | |
|---|---|
| Seed stocks | *Report on the source of all seed stocks or other plant material used. If applicable, state the seed stock centre and catalogue number. If plant specimens were collected from the field, describe the collection location, date and sampling procedures.* |
| Novel plant genotypes | *Describe the methods by which all novel plant genotypes were produced. This includes those generated by transgenic approaches, gene editing, chemical/radiation-based mutagenesis and hybridization. For transgenic lines, describe the transformation method, the number of independent lines analyzed and the generation upon which experiments were performed. For gene-edited lines, describe the editor used, the endogenous sequence targeted for editing, the targeting guide RNA sequence (if applicable) and how the editor was applied.* |
| Authentication | *Describe any authentication procedures for each seed stock used or novel genotype generated. Describe any experiments used to assess the effect of a mutation and, where applicable, how potential secondary effects (e.g. second site T-DNA insertions, mosiacism, off-target gene editing) were examined.* |

